# Multimodal Rehabilitation Management of a Misunderstood Parsonage–Turner Syndrome: A Case Report during the COVID-19 Pandemic

**DOI:** 10.3390/jfmk9010037

**Published:** 2024-02-23

**Authors:** Fabio Santacaterina, Marco Bravi, Mirella Maselli, Federica Bressi, Silvia Sterzi, Sandra Miccinilli

**Affiliations:** 1Department of Physical and Rehabilitation Medicine, Fondazione Policlinico Universitario Campus Bio-Medico, Via Alvaro del Portillo, 200, 00128 Rome, Italy; m.bravi@policlinicocampus.it (M.B.); m.maselli@policlinicocampus.it (M.M.); f.bressi@policlinicocampus.it (F.B.); s.sterzi@policlinicocampus.it (S.S.); s.miccinilli@policlinicocampus.it (S.M.); 2Research Unit of Advanced Robotics and Human-Centred Technologies, Department of Engineering, Università Campus Bio-Medico di Roma, Via Alvaro del Portillo, 21, 00128 Roma, Italy; 3Research Unit of Physical and Rehabilitation Medicine, Department of Medicine and Surgery, Università Campus Bio-Medico di Roma, Via Alvaro del Portillo, 21, 00128 Roma, Italy

**Keywords:** Parsonage–Turner syndrome, brachial plexus neuritis, rehabilitation, shoulder, physiotherapy, referral

## Abstract

During the second wave of the COVID-19 pandemic, a young adult presented symptoms that were reported at first evaluation to be a frozen shoulder (adhesive capsulitis). The patient’s history, clinical manifestations related to the onset of pain, unilateral weakness, and physical examination led to a physiotherapy referral. Subsequent instrumental investigations showed an idiopathic brachial neuritis known as Parsonage–Turner Syndrome (PTS). Contrary to recent descriptions in the literature, the patient did not experience PTS either after COVID-19 vaccination or after COVID-19 virus infection. The proposed multimodal treatment, considering the patient’s characteristics, led to a recovery of muscle strength and function of the upper limb, observed even three years after the acute event. The frequency of rehabilitation treatment, the choice of exercises, the dosage, and the methods of execution require further studies in order to define an evidence-based treatment.

## 1. Introduction

PTS is defined as a peripheral nervous system disorder characterized primarily by two key elements: severe pain and significant muscular atrophy in the shoulder complex [1]. The pain in this unilateral acute plexopathy tends to diminish within a few days or at most two weeks, giving way to marked muscular atrophy, motor paralysis, and sensory deficits [2]. The pathophysiological mechanisms underlying PTS are not entirely clear but appear to involve genetic, immunological, and biomechanical factors [3]. Some authors hypothesize an association between PTS and viral infections, surgery, and vaccinations [3]. A recent systematic review analyzed 26 reported cases of PTS following the administration of the SARS-CoV-2 vaccine [2]. Another systematic review suggested an association not only with COVID-19 infection but also after vaccination, with an earlier onset of PTS symptoms post-vaccination compared to the infection itself [4]. It typically affects adult males more frequently with a sudden onset [5], and sometimes atypical manifestations can occur, such as lower limb and diaphragmatic deficits [6]. The shoulder muscle deficit leads to compensatory movements and scapular dyskinesia, which, while initially useful, can result in a persistent altered pattern and movement limitation at the long term [7]. This aspect is crucial since 60% of patients have residual weakness 2–3 years after the onset of the disease; over 50% are limited by pain; 63% experience severe fatigue; and 82% face difficulties in daily activities [8]. Several strategies have recently been proposed for the rehabilitation of PTS patients, such as a structured multidisciplinary approach that has proven effective [9,10]. This case report shows, for the first time, through an objective strength evaluation system, how recovery can occur even after three years and the role that a progressive multimodal rehabilitation treatment can have in achieving outcomes.

## 2. Materials and Methods

### 2.1. Case Report

This case report, written in line with CARE guidelines [11], describes the diagnostic and rehabilitation process of a 36-year-old Caucasian male, born and raised in Italy, employed in the financial sector, and regularly engaged in physical activities (three gym sessions weekly). He had a history of right wrist fracture surgery (2008), left knee anterior cruciate ligament reconstruction surgery (August 2020), and removal of a lipoma in the left cervical region in 2019. The patient had no comorbidities and was in excellent general health before the acute event.

### 2.2. Clinical Findings

Since December 2020, the patient reported minimal shoulder pain and mild difficulty in right shoulder movement (dominant upper limb). In January 2021, he woke up with severe right shoulder pain (8/10 Numeric Pain Rating Scale—NPRS), unable to raise his arm beyond 90° in active flexion and abduction, as reported from the patient. After an inconclusive shoulder magnetic resonance imaging (MRI), prescribed by a general practitioner on 24 January 2021, he consulted an orthopedic physician in February 2021, who diagnosed “initial adhesive capsulitis of the right shoulder in the context of scapula–humeral girdle mispositioning due to incorrect posture”. Cortisone and hyaluronic acid injections were administered, along with ten individual motor rehabilitation sessions and additional therapies (Tecar^®^ therapy, ultrasound therapy, and low-level laser therapy). Despite the prescribed treatment, the patient showed no improvement after the initial ten sessions and attracted the authors’ attention in April 2021 (Figure 1: case report timeline).

### 2.3. Diagnostic Assessment

A subsequent new physiotherapy assessment on April 2021, through an interview with the patient, observation, and evaluation, revealed scapular alteration [12] (Figure 2A), persistent muscles weakness with marked hypotrophy of the right upper trapezius and sternocleidomastoid, altered shoulder dynamics, and full passive shoulder range of motion (ROM), not found in a typical frozen shoulder 2 months after painful onset, leading to a referral by the physiotherapist to a neurologist in May 2021. The neurologist suspected accessory nerve injury and ordered electromyography (EMG) and cervical plexus MRI. The EMG indicated reduced recruitment with polyphasic potentials and mild fibrillation involving the accessory nerve, long thoracic nerve, and suprascapular nerve. Cervical plexus MRI showed minimal thickening and heterogeneity of the lateral, medial, and posterior cords, suggesting mild neuropathy. The neurologist recommended discontinuing physiotherapy for a month, corticosteroid therapy for one month, and then reassessment. EMG follow-up (June 2021) showed improved nerve conductivity, allowing for the resumption of physiotherapy. The new physiotherapy evaluation, on June 2021, showed an improvement in the performance of the affected muscles, even if still deficient, through muscle evaluation with the Medical Research Council (MRC) (trapezius upper fibers 3/5, serratus anterior 3/5, sternocleidomastoid 4/5) and a reduction in painful symptoms that persisted in the last 20° of active shoulder flexion and last 10° of active shoulder abduction (3/10 NPRS). The patient still reported persistent pain even at rest, especially at night (2/10 NPRS).

### 2.4. Therapeutic Intervention

The resumption of rehabilitation activity in June 2021 took into account the long diagnostic process of the patient who experienced periods of frustration regarding the persistence of muscles weakness and functional deficits. We therefore planned to modulate pain through manual therapy in the short term (Table 1). Furthermore, a triple application of elastic taping, previously used in the literature for reducing shoulder pain and improving function [13], was applied in the short term on a three-weekly basis. At the same time, muscle strengthening activity was resumed through therapeutic exercise to achieve the medium-term objective of recovering muscle strength (Table 1). To this end, we proceeded with the execution of open and closed kinetic chain exercises of the target muscles (Table 1). The exercises were chosen considering the patient’s preferences. The long-term goal was to bring the patient back to resuming his sporting activity (body building) safely and without difficulty (Table 1). An extended description of the treatment protocol is available in Appendix A. The patient was followed until November 2021 with an intensive physiotherapy program. After this date, he continued with home exercises and gym activity on his own. Follow-ups were then carried out up to 3 years after the acute event.

## 3. Results

The patient underwent several follow-ups through physiatry and neurological check-ups to monitor clinical conditions during the rehabilitation period. A new EMG examination (September 2021) reported improvements in the conduction of the affected nervous structures with the persistence of slight signal alterations. A physiatric evaluation (September 2021) also reported clinical improvement with the resolution of pain on movement and at rest (0/10 NPRS), full active shoulder ROM, and the persistence of shoulder muscle strength deficit (trapezius upper fibers 4/5, serratus anterior 4/5, sternocleidomastoid 4/5 MRC), and the need for physiotherapy sessions, performed until November 2021. After the rehabilitation process (12, 24, and 36 months from the acute event), the patient underwent strength tests with a handheld dynamometer (DynaMo, VALD Performance, Australia), with the patient standing upright (Table 2), and administration of the DASH, a patient-reported outcome measure (PROM) (Table 3). The data show how recovery occurred slowly over the 3 year follow-up. Currently the patient has returned to carrying out all the activities of daily life and has reintegrated without problems into his social and working context. He returned to sporting activity in the gym. Furthermore, scapular alteration was no longer appreciable in posterior vision at the December 2023 follow-up (Figure 2B).

## 4. Discussion

PTS is a more prevalent condition than has been previously estimated [14]. A prospective study suggests an incidence of 1 in 1000 [15], confirming that, probably due to its deceptive manifestation, it is underestimated.

The literature indicates that PTS should be considered in diagnostic hypotheses for patients with severe pain and weakness following COVID-19 vaccination [16]. In our specific case, despite occurring during the peak of the second pandemic wave, the patient had not received any vaccine doses and had not contracted the virus. It is our opinion that COVID indirectly influenced a characteristic aspect of this patient: before the physiotherapy referral, which occurred in April 2021, the patient had never been observed without a t-shirt, both in medical evaluations and during physiotherapy treatments. This could be attributed to distancing-related hesitations and fears of possible contagion experienced during the pandemic. An initial lesson from this case report is undoubtedly the importance of observation, inspection, and palpation during the patient’s physical examination, which should always be preceded by a thorough initial conversation. The speed with which some specialist consultations occur, without paying attention to clinical manifestations, such as typical unilateral muscle hypotrophy, scapular alteration, and full passive ROM, can lead to inappropriate therapeutic choices and delay the patient’s recovery. It is essential to consider how PTS can mimic a common shoulder pathology, thus confusing the diagnostic algorithm. It should also be highlighted that the onset with severe pain is common to both PTS [2,5] and frozen shoulder [17,18]; this could therefore have misled the first diagnosis. In fact, the patient, at the first evaluation, reported a high pain level (8/10 NPRS). Furthermore, the scientific literature has already shown how the two pathologies can sometimes be confused [5]. It is also crucial to emphasize the timing in PTS diagnoses; in many cases, denervation is no longer easily identifiable with MRI or EMG after 2–4 weeks [16]. The diagnostic timing in this case report is characteristic, showing that, despite various consultations, a diagnostic hypothesis was only considered after several months. Additionally, clinical presentation, physical examination, and diagnostic tests are useful for differentiating PTS from cervical radiculopathy, thereby reducing healthcare costs and avoiding inappropriate procedures and treatments [19].

The characteristic element of this case report was the proposed rehabilitation treatment; the multimodal approach, related to timing and short-, medium-, and long-term goals, appears to have contributed to the recovery of strength and function of the upper limb. Currently, in the literature, the rehabilitation process lacks certainty, but some elements seem effective and are widely used, such as neurodynamic techniques [20]. Recent studies have shown that a multidisciplinary approach appears valid in managing these patients, emphasizing the importance of sensorimotor recovery, including exercises for regaining upper limb proprioceptive abilities and motor imagery [9,10]. A factor that had a positive influence on the patient’s prognosis in this case report is his young age, motivation, active lifestyle, and adherence to the therapeutic plan (the patient attended all the scheduled sessions). Patient-centered action, the consideration of preferences in exercise selection, and the constant monitoring of conditions are elements that may have contributed to the therapeutic alliance established. An element that may had a role in the recovery of strength was the introduction of modifications to the TUT depending on the specific aims; in fact, it was initially preferred to use a high TUT with low loads, so as to not exacerbate the painful symptoms. Secondly, afterwards, we moved on to increasing the loads by reducing the TUT itself, and, finally, exercises related to the return to sporting activity were carried out. Basing the exercise only on the load and the number of repetitions does not seem to be sufficient for strength recovery, and changes to the TUT may play a role [21], but further studies are necessary, as a recent systematic review has shown that it is still a poorly studied topic, especially regarding the upper limb [22]. The results obtained in the various follow-ups show how the recovery occurred over a very long period and in a progressive manner, both in terms of upper limb strength and functional recovery. Future research should focus on rehabilitation treatment modalities, timing, and exercise specificity to propose evidence-based treatment.

The patient was asked the following questions:

How do you feel about your shoulder today? How do you assess your condition? Do you feel you have returned to the same levels as before? “Currently, I feel better. I don’t think I’ve fully recovered. In the gym, I can do all the exercises, but sometimes I feel like my shoulder ‘slips away’, especially when I’m more tired. Also, when I see myself in the mirror, I notice that my right shoulder is more forward than the other. I feel the strength has returned, but when I’m more tired, I feel I need more control over the body.”

What idea have you formed about what you have experienced in these two years regarding the shoulder disorder? “I think I have to learn to live with these small disturbances that remain. I’m continuing to go to the gym because I believe the condition can still improve”.

## 5. Conclusions

PTS can have an insidious onset and hide behind more common pathologies. Observation, inspection, and physical examination are fundamental elements to evaluate the presence of this condition. Recovery, often not complete, may occur even after three years. The proposed multimodal treatment represents a valid alternative to managing this pathological condition. Further studies are needed to define a fully evidence-based action.

## Figures and Tables

**Figure 1 jfmk-09-00037-f001:**
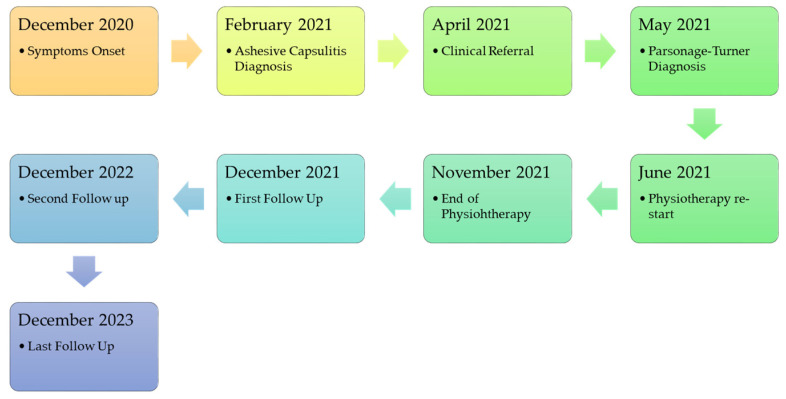
Case report timeline.

**Figure 2 jfmk-09-00037-f002:**
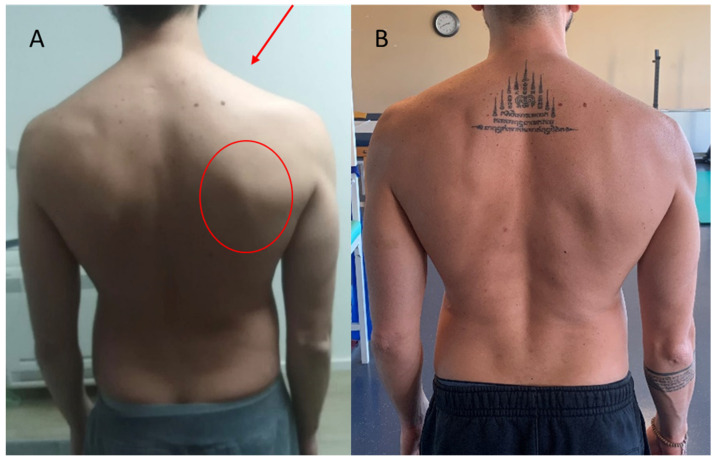
(**A**) Posterior view with scapular inferior angle prominence (red circle). The red arrow shows the upper trapezius hypotrophy, February 2021. (**B**) Posterior view, December 2023.

**Table 1 jfmk-09-00037-t001:** Rehabilitation aims and therapeutic strategies carried out during the rehabilitation process.

Timing	Short Term (within 1 Month)	Mid Term (within 3 Month)	Long Term (within 6 Month)
Aims	Pain modulation and recovery of muscle strength	Complete recovery of muscle strength and movement patterns	Return to sport
Weekly sessions	5 times a week	3 times a week	2 times a week
Treatments	- Myofascial release of the target muscles - Neurodynamic techniques for mechanosensitivity - Shoulder elastic taping - Pompage techniques of the cervical spine, trapezius, sternocleidomastoid, and levator scapulae. - Mobilization with Movement (MWM) for the recovery of the last degrees of flexion and abduction in the absence of pain - Manipulations of the dorsal spine - Therapeutic exercise for shoulder ROM recovery	- Open kinetic chain exercises with progression of intensity, frequency, time under tension, and load parameters - Closed kinetic chain exercises with progression of intensity, frequency, time under tension (TUT), and load parameters - Joint position sense exercise for the upper limb - Exercises using TRX to improve target muscle activation	- Progressions for recovering the movements requested by the patient: push up, pull up, plank, side plank, deadlift, squat

**Table 2 jfmk-09-00037-t002:** Shoulder muscle dynamometer test follow-up results at 12, 24, and 36 months. All force data are normalized to weight (N/Kg).

Dynamometer Shoulder Tests	Affected Side [Right]			Healthy Side [Left]		
	12 Months	24 Months	36 Months	12 Months	24 Months	36 Months
Flexion [N/Kg]	0.88	1.08	1.28	0.95	0.92	1.13
Extension [N/Kg]	2.10	2.18	2.10	1.89	1.93	1.80
Abduction [N/Kg]	0.94	1.09	1.34	1.01	1.10	1.08
Adduction [N/Kg]	1.55	1.86	1.78	1.49	1.62	2.08
Internal rotation [N/Kg]	0.85	0.99	1.03	1.25	1.15	1.19
External rotation [N/Kg]	0.78	0.90	1.03	1.10	1.15	0.99

**Table 3 jfmk-09-00037-t003:** DASH follow-up results at 12, 24, and 36 months for the affected side.

DASH Section	12 Months	24 Months	36 Months
DASH principal (%)	64.2%	45%	21.7%
DASH optional (%)	87.5%	75%	40.6%

## Data Availability

All data relating to quantitative evaluations are reported in the text. For privacy reasons, it is not possible to attach medical reports.

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
