# Peer review of "Multimodal Rehabilitation Management of a Misunderstood Parsonage–Turner Syndrome: A Case Report during the COVID-19 Pandemic"

_jfmk, 2024, doi:10.3390/jfmk9010037_

Round 1
Reviewer 1 Report
Comments and Suggestions for Authors
Dear Authors,
thank you for your efforts in writing the case report titled: "Can Parsonage Turner syndrome masquerade as Frozen Shoulder Contracture syndrome? a case report during the COVID-19 pandemic."
In my opinion, there was no motive to write this case report: no novelty, no alternative rehabilitation treatment, no differential diagnosis path. I would suggest to Authors some papers for better understand a general aim of a case report, that is not only t describe a un useful case (as in this case), but o make a substantial contribution to the body of knowledge in a specific area of clinical practice (1- Neely JG, Karni RJ, Nussenbaum B, Paniello RC, Fraley PL, Wang EW, Rich JT. Practical guide to understanding the value of case reports. Otolaryngol Head Neck Surg. 2008 Mar;138(3):261-4. 2- Aitken LM, Marshall AP. Writing a case study: Ensuring a meaningful contribution to the literature. Aust Crit Care. 2007 Nov;20(4):132-6. Epub 2007 Oct 23. 3- Malay DS. The value of an interesting case. J Foot Ankle Surg. 2007 Jul-Aug; 46(4): 2112.).
CARE guideline are a methodological and reporting guideline for case report. the paragraph in lines 47-48 should be moved in Method section.
Lines49-50, "initially confused with frozen shoulder contracture syndrome". this sentence was not thorough in the manuscript and had any importance in the case described by the Authors. Moreover, I think is impossible to mistake Frozen shoulder with PTS, as the first one is characterised by a anctive and passive movement restriction, while in the latter only acrive rstriction was present.
Manuscript should be written in the "past" (line 54)
Line 74: figure 2 cames before figure 1
Line 77: authors should explode the Acronyms the first time that they use them
Figure 1: timeline lacks of follow up
Line 93: how did the Authors able to select assess the supraspinatus by MRC? I think is impossible
Line 93: Serratus anterior and not Major
Line 94: no specific information about range of motion restriction was given. "last grades" is not infomrative
Line 104: end of physiotherapy (nov 2021) was not present in the timeline
Table 1. Therapeutic plan, that was the core of the Manuscript, was not described in deep, not detailed and not photograpfs were showed. This is a major concern, in my opinion
Line 109: "constant follow ups" Could the Authors give further specifications?
Line 115: suggest to delete "clinical". DASH is patient-reported, not clinical
Line 120: "Right side is the affected and the dominant side" is redundant and indicated into the table. I suggest to delete it.
FIgure 2: In my opinion Figure 2A showed an inferior angle prominence (tipe 1 diskinesia by Kibler et al.) and not a scapular winging.
PLease describe in the Figure 2 note the means of the arrow in figure 2A
Line 158: Authors wrote about patient's adherece to therapeutic plan, but no information was provided. how did they assessed it?
Line 166: Authors wrote that the patinet was asked 2 qustions, but the interrogative dots were 4
Lines 183-184 Authors wrote "multidisciplinary approach with a multimodal treatment represents the essential key to managing this pathological condition" but no multidisciplinar treatment was described in the manuscript
Comments on the Quality of English Language
moderate editing of english langiage should be requested to an English mother language
Author Response
Reviewer 1
Comments and Suggestions for Authors
Dear Authors,
thank you for your efforts in writing the case report titled: "Can Parsonage Turner syndrome masquerade as Frozen Shoulder Contracture syndrome? a case report during the COVID-19 pandemic."
In my opinion, there was no motive to write this case report: no novelty, no alternative rehabilitation treatment, no differential diagnosis path. I would suggest to Authors some papers for better understand a general aim of a case report, that is not only t describe a un useful case (as in this case), but o make a substantial contribution to the body of knowledge in a specific area of clinical practice (1- Neely JG, Karni RJ, Nussenbaum B, Paniello RC, Fraley PL, Wang EW, Rich JT. Practical guide to understanding the value of case reports. Otolaryngol Head Neck Surg. 2008 Mar;138(3):261-4. 2- Aitken LM, Marshall AP. Writing a case study: Ensuring a meaningful contribution to the literature. Aust Crit Care. 2007 Nov;20(4):132-6. Epub 2007 Oct 23. 3- Malay DS. The value of an interesting case. J Foot Ankle Surg. 2007 Jul-Aug; 46(4): 2112.).
We thank the reviewer for the valuable advice that will lead to a scientific article of greater impact. Compared to the lack of novelty in the case report presented, we tried to modify the main objective, we no longer focused on the differential diagnosis but on the rehabilitation treatment: in fact, currently there are no guidelines regarding the rehabilitation treatment methods, no case report enters the specific to the exercises performed, the dosage and the methods chosen. Furthermore, compared to what has been found in the literature, this is the first case in which strength recovery is analyzed through an objective measuring instrument (traction dynamometer), offering a follow-up of up to three years.
To this end, the title and some elements within the article have been changed, and the description of the rehabilitation treatment has had greater emphasis than the diagnostic classification. A complete description of the rehabilitation protocol has been created in Appendix A.
CARE guideline are a methodological and reporting guideline for case report. the paragraph in lines 47-48 should be moved in Method section.
Thanks for the comment, the reference has been placed in the materials and methods section as requested.
Lines49-50, "initially confused with frozen shoulder contracture syndrome". this sentence was not thorough in the manuscript and had any importance in the case described by the Authors. Moreover, I think is impossible to mistake Frozen shoulder with PTS, as the first one is characterised by a anctive and passive movement restriction, while in the latter only acrive rstriction was present.
Thanks for the comment. One of the reasons that pushed us to describe this case report, in addition to the introduction of objective systems for monitoring the recovery of strength and specific rehabilitation techniques, was precisely because we also believed it was "impossible" to confuse these two pathologies.
However, the literature does not agree with our first impression nor with the reviewer, as adhesive capsulitis (or frozen shoulder contracture syndrome) is considered to all intents and purposes to be a pathology that can be differentially diagnosed with Turner parsonage, and is even considered the first pathology with which it is most confused ( Feinberg JH, Radecki J. Parsonage-turner syndrome. HSS J. 2010 Sep;6(2):199-205. doi: 10.1007/s11420-010-9176-x. Epub 2010 Jul 30. PMID: 21886536; PMCID: PMC2926354).
From Feinberg et al 2010: “Several similarities exist between Parsonage-Turner Syndrome and adhesive capsulitis. Both conditions present with severe pain, worse at night, and, initially, are unremitting regardless of position. Both conditions are idiopathic, have a nonspecific inflammatory component, and will resolve spontaneously with a relatively good long term prognosis for recovery of function”
In reality this happened, in our opinion, as described in the "Discussion "It is our opinion that COVID indirectly influenced a characteristic aspect of this patient: before the physiotherapy referral, occurred in April 2021, the patient had never been observed without t-shirt, both in medical evaluations and during physiotherapy treatments."
In fact, the patient told us that in the previous two visits (one orthopedic and the other physiatrists) and during all the first 10 rehabilitation sessions with two different physiotherapists, no one took his shirt off but he was always examined through passive mobilization and a request for active movements always wearing a t-shirt. We have added this detail to clarify the clinical case in the DISCUSSION session. We agree with the reviewer that obviously in the case of Frozen shoulder Contracture syndrome the passive range is limited (even if this is not always present in the very early stages of the pathology as reported from Lewis et al 2015, Chan et al 2017…) and that therefore this could be a strong indication towards another pathology.
However we belive that the patient's severe pain (8/10 NPRS) led the previous doctors down the wrong path and above all, due to the period of social distancing and a certain resistance of some clinicians to conducting in-depth objective examinations, it may have had an impact on the patient's diagnosis. We also clarified that point in the discussion part “It should also be underlined that the onset with severe pain is common to both PTS [2,5] and Frozen Shoulder contracture syndrome [16,17], this could therefore have misled the doctors themselves”.
Figure 1 from Chan, H.B.Y.; Pua, P.Y.; How, C.H. Physical Therapy in the Management of Frozen Shoulder. Singapore Med J 2017, 58, 685–689, doi:10.11622/SMEDJ.2017107.
Our warning therefore was to consider this pathology but above all to conduct a complete objective examination because, as in this case, it can be decisive. Could it be obvious? We thought so too before finding ourselves faced with a situation like this, so we think it is necessary to reinforce the message.
The first thing we noticed when the patient came to our attention in April 2021, in addition to the complete passive ROM and the scapulothoracic alteration, was the marked hypotrophy of the upper trapezius fibres of the affected side compared to the healthy one.
This item was specified in the article. The previous doctors may not have initially had the opportunity to notice this detail but, subsequently, the physiotherapists should also have undressed the patient and had the opportunity to notice the marked hypotrophy that was created over time and the full passive ROM.
Manuscript should be written in the "past" (line 54)
Thanks for the comment, we changed the entire period.
Line 74: figure 2 cames before figure 1
Thanks for the comment, we corrected the manuscript in order to have figure 1 first.
Line 77: authors should explode the Acronyms the first time that they use them
Thanks for the comment, we corrected.
Figure 1: timeline lacks of follow up
Thanks for the comment, we modified as requested.
Line 93: how did the Authors able to select assess the supraspinatus by MRC? I think is impossible
Thanks for the comment, we agree with the reviewer, the delete this evaluation.
Line 93: Serratus anterior and not Major
Thanks for the comment, we changed as requested.
Line 94: no specific information about range of motion restriction was given. "last grades" is not informative
Thanks for the comment, we modified as requested.
Line 104: end of physiotherapy (nov 2021) was not present in the timeline
Thanks for the comment, we modified as requested.
Table 1. Therapeutic plan, that was the core of the Manuscript, was not described in deep, not detailed and not photograpfs were showed. This is a major concern, in my opinion
Thanks for the comment, a detailed description (appendix A) of the protocol administered to the patient has been added, also through the use of photos.
Line 109: "constant follow ups" Could the Authors give further specifications?
Thanks for the comment, we modified as requested.
Line 115: suggest to delete "clinical". DASH is patient-reported, not clinical
Thanks for the comment, we modified as requested.
Line 120: "Right side is the affected and the dominant side" is redundant and indicated into the table. I suggest to delete it.
Thanks for the comment, we removed as requested.
FIgure 2: In my opinion Figure 2A showed an inferior angle prominence (tipe 1 diskinesia by Kibler et al.) and not a scapular winging.
Thanks for the comment, we modified as requested.
PLease describe in the Figure 2 note the means of the arrow in figure 2A
Thanks for the comment, we modified as requested.
Line 158: Authors wrote about patient's adherece to therapeutic plan, but no information was provided. how did they assessed it?
Thanks for the comment, we implemented in the text.
Line 166: Authors wrote that the patinet was asked 2 qustions, but the interrogative dots were 4
Thanks for the comment, we modified as requested.
Lines 183-184 Authors wrote "multidisciplinary approach with a multimodal treatment represents the essential key to managing this pathological condition" but no multidisciplinar treatment was described in the manuscript
Thanks for the comment, we modified as requested.
Reviewer 2 Report
Comments and Suggestions for Authors
I would like to thank the editor for the opportunity to review this manuscript. The manuscript presents a case report of a comprehensive rehabilitation approach of a young adult with Parsonage-Turner Syndrome. Its findings may be of great clinical importance for clinicians encountering individuals with this pathology in clinical settings. Nevertheless, I have some major concerns, that should be carefully addressed prior to considering this manuscript for publication. Additionally, the manuscript requires proof reading, particularly regarding syntax and grammar.
Major comments:
- The title of the manuscript warrants re-evaluation and modification. The main topic of the manuscript is rehabilitation of Parsonage-Turner syndrome and not differential diagnosis as the title suggests. Although I agree, that this interesting case highlights the importance of accurate diagnosis for enabling subsequent effective treatment, the diagnosis was not an important aspect of the empirical part of this case report, therefore the title and the manuscript should be adjusted accordingly.
- In line with the previous comment, I appeal to the authors to modify the Abstract by including more of the rehabilitation and less diagnosis aspects. Specifically, line 19-21 is not related to the main purpose of this case report and should therefore be removed/modified.
- The methods section should include detailed description of the methodology for assessment of the included outcome measures.
- The discussion should be more focused on the empirical findings of the study and therefore requires extensive revision.
Minor comments:
- Line 17-19: I recommend modifying the order of words: »Contrary to recent descriptions in the literature, the patient did not experience PTS […] «.
- Line 34 and 37: I recommend replacing the term »correlation« with »association«.
- Line 67: Please change »injection« to »injections«.
- Line 91: I suggest replacing the term »recruitment capacity« with »performance«, as the former relates more to the activation (neural) components of muscle performance, which were not directly assessed with the described tests.
- Line 94: It would be beneficial fort the readers interpretation to add specific information regarding the type of movement, e.g. active or passive.
- Line 102: I recommend changing the term »created« with »chosen«.
- Line 103: I suggest the authors include information regarding the participants goal sporting activity, as this specific information is missing.
- Table 1: To my knowledge the term »pumping techniques« is not a general term and should be explained or changed for the benefit of clear understanding.
- Line 114: Please consider replacing the term »static« with »upright«.
- Table 2: If possible, consider providing normalized torque values for shoulder muscle strength. This would enable direct comparisons with reference values.
- Line 182-183: The term »can take place« should be replaced with a more appropriate term.
Comments on the Quality of English Language
The manuscript requires proof reading, particularly regarding syntax and grammar.
Author Response
Major comments:
- The title of the manuscript warrants re-evaluation and modification. The main topic of the manuscript is rehabilitation of Parsonage-Turner syndrome and not differential diagnosis as the title suggests. Although I agree, that this interesting case highlights the importance of accurate diagnosis for enabling subsequent effective treatment, the diagnosis was not an important aspect of the empirical part of this case report, therefore the title and the manuscript should be adjusted accordingly.
Thank you for your valuable comment, we have changed the title as requested. We have also implemented the discussion by focusing on the rehabilitation protocol, substantially modified some parts and added an appendix A to clarify all the exercises, also through photos.
- In line with the previous comment, I appeal to the authors to modify the Abstract by including more of the rehabilitation and less diagnosis aspects. Specifically, line 19-21 is not related to the main purpose of this case report and should therefore be removed/modified.
Thanks for the comment, we deleted as requested.
- The methods section should include detailed description of the methodology for assessment of the included outcome measures.
Thanks for the comment, we have tried to specify the evaluation methods within the entire article but, in accordance with what is specified in the CARE guidelines, we have avoided creating a section of materials and methods that tells a priori the choice of the outcomes evaluated, and we have given greater emphasis on the temporal sequence of events, detailing as much as possible the tools used for the evaluation and the results obtained. However, we have moved the June 2021 physiotherapy assessment described in therapeutic intervention to the diagnostic assessment section for greater clarity.
- The discussion should be more focused on the empirical findings of the study and therefore requires extensive revision.
Thanks for the comment. The discussion section has been implemented, providing more emphasis on rehabilitation treatment.
Minor comments:
- Line 17-19: I recommend modifying the order of words: »Contrary to recent descriptions in the literature, the patient did not experience PTS […] «.
Thanks for the comment, we modified as requested.
- Line 34 and 37: I recommend replacing the term »correlation« with »association«.
Thanks for the comment, we modified as requested.
- Line 67: Please change »injection« to »injections«.
Thanks for the comment, we modified as requested.
- Line 91: I suggest replacing the term »recruitment capacity« with »performance«, as the former relates more to the activation (neural) components of muscle performance, which were not directly assessed with the described tests.
Thanks for the comment, we modified as requested.
- Line 94: It would be beneficial fort the readers interpretation to add specific information regarding the type of movement, e.g. active or passive.
Thanks for the comment, we modified as requested.
- Line 102: I recommend changing the term »created« with »chosen«.
Thanks for the comment, we modified as requested.
- Line 103: I suggest the authors include information regarding the participants goal sporting activity, as this specific information is missing.
Thanks for the comment, we included the information as requested.
- Table 1: To my knowledge the term »pumping techniques« is not a general term and should be explained or changed for the benefit of clear understanding.
Thanks for the comment, we modified as requested with “pompage”.
- Line 114: Please consider replacing the term »static« with »upright«.
Thanks for the comment, we modified as requested.
- Table 2: If possible, consider providing normalized torque values for shoulder muscle strength. This would enable direct comparisons with reference values
Thanks for the comment. We don't understand this request, can you be more specific? What is meant by normalizing the data? The data are actually normalized as the formula for normalizing strength with respect to the patient's body weight has been applied, as described in the table. Thank you-
- Line 182-183: The term »can take place« should be replaced with a more appropriate term.
Thanks for the comment. We modified es requested.
Round 2
Reviewer 1 Report
Comments and Suggestions for Authors
I commend to the Authors for the efforts in improving the Manuscript. However, some other amendments were required:
TITLE: I suggest to delete “misunderstood” as the manuscript focused on the rehabilitative approach for PTS
ABSTRACT: As above in the title. Writing about frozen shoulder defocused the rearder, I suggest to remove. This is not a case report that investigated abboout differential diagnosis, but on rehabilitative approach
KEYWORDS: please ensure that all keyword were MEsH terms
LINE 86: please change RMI to MRI
LINE 93: please change “our” with “authors’”
LINE 98: please delete “Type 1 dyskinetic scapular pattern” (see the figure 2)
LINE 122: please delete “contracture syndrome”. These specifications are already questioned because of the
psychosocial load on “syndrome” and the uncertainty about the cause of the movement restrictions (muscular, connective tissues, or both. It is simply Frozen Shoulder. Also in line 242
LINE 143-144: The “self-citated” study number 13 is QUITE THE ONLY study in literature that supported the effectiveness of the elastic tpaing in subjects with shoulder pain (see: doi: 10.1016/j.physio.2019.12.001; Cochrane Database Syst Rev. 2021 Aug 8;8(8):CD012720. doi: 10.1002/14651858.CD012720.pub2), i suggest to delete.
FIGURE 2. Please change “Posterior view with scapular inferior medial border prominence, type 1 scapula dyskinetic pattern (red circle)” February 2021. B) Posterior view December 2023. Red arrow shows the 220 upper trapezius hypotrophy.” To “Posterior view with scapular inferior angle prominence (red circle)”. Red arrow shows the upper trapezius hypotrophy. February 2021. B) Posterior view December 2023.
It can not be “dyskinesis” because the subject did not move. It is a position!
Please be consistent with acronyms (RMN, PTS…) throughout the text
APPENDIX A
LINE 345: please change “explain” to “show/report”
I suggest to add active exercises reports, instead of passive therapeutic approaches
Please be consistent with acronyms throughout the Appendix
Comments on the Quality of English LanguageI m not qualified to review the quality of English
Author Response
Comments and Suggestions for Authors
I commend to the Authors for the efforts in improving the Manuscript. However, some other amendments were required:
TITLE: I suggest to delete “misunderstood” as the manuscript focused on the rehabilitative approach for PTS
We thank the reviewer for the comment. We believe that the emphasis we propose both in the title and in the abstract regarding the first diagnosis is not confusing for the reader as it is strictly connected with the rehabilitation decisions that have been made: the physiotherapy referral is an integral part of the rehabilitation project that has been described and, in our opinion, deserves the reader's attention.
This article focuses on the rehabilitation of a patient with PTS: the rehabilitation process consists of an evaluation phase, a goal setting phase, a treatment phase, and constant follow-ups to understand if the treatment set is truly consistent with the set aims. Canceling screening for referral means, in our opinion, not considering the main goal of the physiotherapist, and of any healthcare professional in general: to be sure of moving within their own scope of care.
We have shown how the literature agrees that the delay in diagnosis can negatively influence the outcomes, and how we have taken this into account when setting up the rehabilitation project itself: this element was perceived as frustrating for the patient and the rehabilitation protocol sought to adherence of the patient himself who was distrustful of the healthcare world and worried about his own health status.
We have made the necessary changes to make the paper as requested, omitting the fact that there was an incorrect initial classification in our opinion would not justify some actions undertaken in the construction of the rehabilitation project. It is not so much a question of diagnosis but of screening for referral, in fact we believe, as already expressed in the article, that at the beginning the two pathologies can be confused, as is WIDELY demonstrated in the literature and we expressed in ROUND 1. The real problem it is precisely that the first physiotherapists who took care of the patient should have implemented the screening for referral tool and understood that the patient at that stage was outside their scope of treatment and needed medical intervention.
ABSTRACT: As above in the title. Writing about frozen shoulder defocused the rearder, I suggest to remove. This is not a case report that investigated abboout differential diagnosis, but on rehabilitative approach
Thanks for the comment. What we expressed in the first comment also applies to what was requested in the abstract: we have modified by eliminating the word "contracture syndrome", but we believe that adding this aspect also in the abstract is not only not confusing, but helps the reader himself to understand better and frame the case report and the subsequent rehabilitation protocol adopted, which is the very center of the case report.
The abstract has already been substantially changed as requested in the first round and it is clear that the core of the article is the rehabilitation approach, as can easily be understood from the title itself.
KEYWORDS: please ensure that all keyword were MEsH terms
Thanks for the comment. We changed one term as it didn't appear to be a Mesh term.
LINE 86: please change RMI to MRI
Thanks for the comment. We have modified as requested.
LINE 93: please change “our” with “authors’”
Thanks for the comment. We have modified as requested.
LINE 98: please delete “Type 1 dyskinetic scapular pattern” (see the figure 2)
Thanks for the comment, we modified as requested. Please refer to our comment to yours at figure 2.
LINE 122: please delete “contracture syndrome”. These specifications are already questioned because of the psychosocial load on “syndrome” and the uncertainty about the cause of the movement restrictions (muscular, connective tissues, or both. It is simply Frozen Shoulder. Also in line 242
Thanks for the comment, we modified as requested.
LINE 143-144: The “self-citated” study number 13 is QUITE THE ONLY study in literature that supported the effectiveness of the elastic tpaing in subjects with shoulder pain (see: doi: 10.1016/j.physio.2019.12.001; Cochrane Database Syst Rev. 2021 Aug 8;8(8):CD012720. doi: 10.1002/14651858.CD012720.pub2), i suggest to delete.
thanks for the comment. We do not understand this request: the article was cited only in order to explain how the taping is applied. In fact, in the article cited, an application technique is shown that brought short-term results in the modulation of pain and in the recovery of function and ROM, the same objectives that we had with the patient in the case report. The systematic review cited by the reviewer does nothing but confirm what is widely expressed in the literature: elastic taping appears to have no effects on rehabilitation outcomes. But the MAIN limitation of the studies is that the same application methods are not used for the same pathology treated and outcomes detected. In fact, opening the systematic review cited by the reviewer shows that the application methodology is heterogeneous. So we don't understand why the citation should be eliminated. If we eliminated it, we would have to describe the same technique we used in the study, with the same weekly dosage, the same tension applied, etc. At this point it might be more consistent to request that you delete that elastic taping was used and then delete the quote itself. But why should we delete it if we have used it? We would like to clarify that the article was never self-cited, if we did so in this case it is only because the same technique was used, among other things described for the first time, with these methods, by the same authors, to obtain from results consistent with the patient's clinical condition.
It can be agreed that ours is among the few studies that supports the effectiveness of the use of elastic taping, but in combination with the use of therapeutic exercise, and not as the only strategy. This is also what was done in the case report. And above all in the other studies similar, but not identical, techniques are used. If there were studies showing the inconsistency of the same technique, it is not within our knowledge. We therefore kindly ask the reviewer to be able to access this information so, at that point, we can state that the technique has been used, but that only one study in the literature supports it while others have not obtained results. However we deleted “successfully” when we talked about elastic taping.
FIGURE 2. Please change “Posterior view with scapular inferior medial border prominence, type 1 scapula dyskinetic pattern (red circle)” February 2021. B) Posterior view December 2023. Red arrow shows the 220 upper trapezius hypotrophy.” To “Posterior view with scapular inferior angle prominence (red circle)”. Red arrow shows the upper trapezius hypotrophy. February 2021. B) Posterior view December 2023.
It can not be “dyskinesis” because the subject did not move. It is a position!
Thanks for the comment. We have modified as requested. Collaterally we would like to point out to the reviewer that this change was made based on what he reported in ROUND 1 “in my opinion Figure 2A showed an inferior angle prominence (type 1 diskinesia by kibler et al.)”.
We used this article: Kibler WB, Uhl TL, Maddux JW, Brooks PV, Zeller B, McMullen J. Qualitative clinical evaluation of scapular dysfunction: a reliability study. J Shoulder Elbow Surg. 2002 Nov-Dec;11(6):550-6. doi: 10.1067/mse.2002.126766. PMID: 12469078.
We referred to the type I scapular dyskinesis system described in Kibler 2002. However we modified as requested. He is Kibler that call it DYSKINETIC SCAPULAR PATTERN TYPE 1 (SEE FIGURE 1 kibler 2002)
Please be consistent with acronyms (RMN, PTS…) throughout the text
Thanks for the comment, we have checked and changed.
APPENDIX A
LINE 345: please change “explain” to “show/report”
Thanks for the comment, we modified as requested.
I suggest to add active exercises reports, instead of passive therapeutic approaches
Thanks for the comment, we added as requested.
Please be consistent with acronyms throughout the Appendix
Thanks for the comment, we have checked and changed.
